# Characterization of Burn Eschar Pericytes

**DOI:** 10.3390/jcm9020606

**Published:** 2020-02-24

**Authors:** Alexander Evdokiou, Onur Kanisicak, Stephanie Gierek, Amanda Barry, Malina J. Ivey, Xiang Zhang, Richard J. Bodnar, Latha Satish

**Affiliations:** 1Shriners Hospitals for Children, Research Department, Cincinnati, OH 45229, USA; evdokioua@gmail.com (A.E.); sjgierek@gmail.com (S.G.); ambarry@shrinenet.org (A.B.); 2Department of Pathology and Laboratory Medicine, University of Cincinnati, Cincinnati, OH 45267-0529, USA; kanisior@ucmail.uc.edu (O.K.); malina.ivey@uc.edu (M.J.I.); 3Genomics, Epigenomics and Sequencing Core, University of Cincinnati, Cincinnati, OH 45267, USA; zhanx5@ucmail.uc.edu; 4Veterans Affairs Medical Center, University Dr. C, Pittsburgh, PA 15240, USA; Richard.bodnar@va.gov

**Keywords:** pericyte, periostin, FoxE1, IL-6, IL-8, MYD88, TGF-β, endosialin, burns and stem cells

## Abstract

Pericytes are cells that reside adjacent to microvasculature and regulate vascular function. Pericytes gained great interest in the field of wound healing and regenerative medicine due to their multipotential fate and ability to enhance angiogenesis. In burn wounds, scarring and scar contractures are the major pathologic feature and cause loss of mobility. The present study investigated the influence of burn wound environment on pericytes during wound healing. Pericytes isolated from normal skin and tangentially excised burn eschar tissues were analyzed for differences in gene and protein expression using RNA-seq., immunocytochemistry, and ELISA analyses. RNA-seq identified 443 differentially expressed genes between normal- and burn eschar-derived pericytes. Whereas, comparing normal skin pericytes to normal skin fibroblasts identified 1021 distinct genes and comparing burn eschar pericytes to normal skin fibroblasts identified 2449 differential genes. Altogether, forkhead box E1 (FOXE1), a transcription factor, was identified as a unique marker for skin pericytes. Interestingly, FOXE1 levels were significantly elevated in burn eschar pericytes compared to normal. Additionally, burn wound pericytes showed increased expression of profibrotic genes periostin, fibronectin, and endosialin and a gain in contractile function, suggesting a contribution to scarring and fibrosis. Our findings suggest that the burn wound environment promotes pericytes to differentiate into a myofibroblast-like phenotype promoting scar formation and fibrosis.

## 1. Introduction

Severe burn injury results in devastating scarring and contractures. Several cell populations, including inflammatory cells, fibroblasts, and keratinocytes, play a crucial role in coordinating the wound healing outcome [1,2]. Recently, pericytes (PCs), a mesenchymal stem cell-like population, have been reported to contribute to liver [3], lung [4], kidney [5], and skin fibrosis [6]. However, PCs role in burn wound healing is still yet to be deciphered. Burn wound environment is an atypical niche compared to incisional or excisional skin injury with excessive inflammation and a prolonged influx of inflammatory cells lasting for several days [7]. The goal of the present study is to capture the changes that might occur to pericytes in burn wounds and their role in wound healing including scarring and contractures.

PCs are perivascular cells with long surface processes that encircle microvascular endothelial cells (ECs) and provide important structural integrity to the vessel wall through direct and indirect signaling mechanisms. [8,9]. PCs promote activation of latent transforming growth factor (TGF)-β inducing EC proliferation and differentiation through TβR/ALK/Smad pathways [10], regulate vessel remodeling through the IP-10/CXCR3 signaling pathway [11,12] and maturation through the expression of Ang 1 [13]. In addition, PCs can regulate immune response through the secretion of cytokines and chemokines [14,15].

Due to limited cell-specific markers, identification and characterization of PCs in burn wound healing have been unreliable. Some of the commonly used markers to identify PCs are neuron-glial 2 (NG2), platelet-derived growth factor receptor beta (PDGFR-β), CD146, alpha-smooth muscle actin (α-SMA), and regulator of G protein signaling 5 (RGS-5) [16]. Mesenchymal stem cell (MSC) markers CD44, CD73, CD90, and CD105 have also been used to identify pericytes [16]. During vessel remodeling however, the expression of CD146 and PDGFR- seem to be ubiquitous, and RGS-5 is upregulated on activated PCs [17]. Pericytes have been observed to be multipotent, thus have been identified as a MSC progenitor [16,18,19]. It is well established that MSC site of origin influences their functional properties, pericyte function seem to also be influenced by their origin and surroundings [20]. It has been postulated that pericyte function may be different in fetal/young tissue compared to adult tissue and not all pericytes are MSCs [21]. Therefore, understanding the signals that regulate pericyte function will provide a better understanding of the role pericytes play in tissue regeneration.

Recent studies have shown that pericytes play a role in wound healing [22,23,24,25], but their significance is not well understood. In wound healing, pericytes contribute to angiogenesis (EC migration and proliferation, wound stabilization, and regression) [12,26,27], inflammation (neutrophil and macrophage extravasation, and T cell activation) [28,29,30,31], re-epithelialization [32], tissue regeneration [29], and fibrosis [32,33]. Our understanding of the involvement of PCs in these functions is still in its infancy. The inflammatory phase of wound healing plays a critical role in the healing process. Any perturbations in the process can lead to delayed healing, fibrosis, or ulcer formation. In this phase, PCs have been observed to have multiple functions. They help to initiate angiogenesis by retracting from the microvessels, promoting dissociation of cell-cell interaction, and promoting EC migration and proliferation [25]. PCs have also been found to be directly involved in the extravasation of neutrophils and macrophages [34]. In addition, matrix deposition in wound healing is essential for wound closure and strength of the skin. Fibroblasts are the principal cell involved in matrix deposition and organization. The dysregulation of fibroblasts promotes excessive collagen production which can lead to fibrosis. Recent studies have shown that PCs have the ability to secrete collagen and contribute to fibrosis [35,36,37]. In a study by Birbrair et al., two distinct PC sub-types were identified, where type 1 PCs secrete collagen and promote fibrosis [35]. These studies suggest that dysregulation of pericytes can play a significant role in fibrosis. Here we evaluated the gene expression of pericytes from burn eschar. We are the first to identify the expression of forkhead box E1 (FOXE1) in pericytes. We also show that pericytes from burn eschar have a high expression of the pro-fibrotic genes fibronectin, myeloid differentiation factor 88 (MyD88), periostin and endosialin, and the pro-inflammatory cytokines IL-6 and IL-8. Characterization of pericytes from burn eschar suggest that the burn wound environment potentiates pericytes to a myofibroblast-like phenotype (collagen producing cell), which can lead to or promote fibrosis.

## 2. Experimental Section

### 2.1. Discarded Human Skin Tissues

Tangentially excised burn eschar was obtained from the routine debridement procedure, which occurs within a week after burn injury. Healthy normal skin was obtained as excess during elective plastic surgery procedures. All tissue was obtained from patients treated at Shriners Hospitals for Children-Cincinnati or the University of Cincinnati Medical Center, Cincinnati, OH. The University of Cincinnati Institutional Review Board determined that this does not constitute human subjects research and is exempt from requirements for informed consent according to 45CFR46.101 (b) (4). Patient information was anonymized, and samples were de-identified prior to analysis. Strain numbers were used to enable de-identification and were assigned sequentially to all the samples collected by the laboratory. For the current study, experiments were performed using burn eschar tissues collected from six different patients, and excess discarded normal skin from six different patients were used. Table 1 provides demographic information.

### 2.2. Pericytes and Fibroblasts Cultures

Primary cultures of pericytes were isolated from discarded normal and burn eschar skin without the use of enzymatic digestion. Skin samples were placed into Pericyte Medium (Zenbio Inc. Research Triangle, NC, USA), cut into thin strips, and scrapped using forceps. Following a 48 h incubation period, the dish was washed twice with 1× PBS (Thermo Fisher Scientific, Waltham, MA, USA), and the media was changed. The cells were allowed to expand to 80% confluence and then frozen for future analysis. Primary cultures of fibroblasts established using discarded normal skin was used as described in detail elsewhere [38].

### 2.3. Flow Cytometry

A Beckman Coulter MoFlo Astrios EQ flow cytometer (Beckman Coulter, Indianapolis, IN, USA) was used to identify and enrich the pericyte population from the cells isolated from the eschar skin using the following panel of antibodies: positive PE conjugated anti-CD146 (BD Biosciences, San Jose, CA, USA), positive APC conjugated CD-105 (BioLegend, San Diego, CA, USA), positive Brilliant Violet 421 conjugated anti-CD73 (BioLegend), negative SuperBright 600 conjugated anti-CD56 (Thermo Fisher Scientific, Waltham, MA, USA), negative BrilliantBlue 515 conjugated anti-CD45 (BD Biosciences), and negative PerCP-eFluor 710 conjugated anti-CD34 (Thermo Fisher Scientific). Sorted pericytes were used for each experiment. A BD LSRII flow cytometer (BD Biosciences) was used to phenotype the expression of surface markers on pericytes and fibroblasts using PE conjugated anti-CD146 (BD Biosciences), APC conjugated CD-105 (BioLegend), Brilliant Violet 421 conjugated anti-CD73 (BioLegend), SuperBright 600 conjugated anti-CD56 (Thermo Fisher Scientific), BrilliantBlue 515 conjugated anti-CD45 (BD Biosciences), and PerCP-eFluor 710 conjugated anti-CD34 (Thermo Fisher Scientific). This panel was specifically chosen using the application Fluoro Finder to minimize spectral overlap. 1 × 10^6^ cells were placed in individual polystyrene tubes, washed twice with 1× PBS, fixed using 4% paraformaldehyde (Affymetrix; Thermo Fisher Scientific) for 10 min, washed twice more, then stained for marker expression. During analysis, 10,000 events per antibody of interest were analyzed, and data are presented as % positive of the gated population ± SEM.

### 2.4. Immunocytochemistry & Immunohistochemistry

Immunocytochemistry: Pericytes and fibroblasts were grown to 85% confluence on 8-well chamber polystyrene slides (Falcon; Thermo Fisher Scientific) at a concentration of 10^4^ cells per chamber. Cells were fixed with 4% paraformaldehyde and washed twice with 1× PBS. Following fixation, cells were stained with anti-CD146 (1:250; Abcam, Cambridge, MA, USA), S100A4 (1:250; Abcam), NG2 (1:200 Sigma-Aldrich, St. Louis, MO, USA), myosin smooth muscle heavy chain (MHY11, 1:200; Invitrogen, Thermo Fisher Scientific; MA511971), Calponin-1 (1:200; Invitrogen MA511620), Transgelin/ smooth muscle 22 (SM22) (1:200; Proteintech 104931AP) and incubated overnight at 4 °C. Slides were then washed twice with 1× PBS, stained with DAPI mounting medium (Vector Laboratories Inc, Burlingame, CA, USA), and imaged using Eclipse 90i microscope equipped with a Ds-Ri1 Digital Microscope Camera (Nikon Instruments Inc., Melville, NY, USA).

Immunohistochemistry: Discarded normal skin and burn eschar tissue were embedded frozen for cryosectioning using OCT Compound (Fisher HealthCare, Pittsburgh, PA, USA). Immunohistochemistry was performed on cryosections prepared by Shriners Hospitals for Children-Cincinnati Histology Special Shared Facility. Sections were incubated with primary antibody for CD-146 (cat.no. ab 78451; 1:200; Abcam) and periostin (cat. no: ab 215199; 1:200; Abcam) overnight at 4 °C. Secondary antibodies against CD-146 (ab 150117; Abcam; 1:400) and periostin (A21207; Thermo Fisher Scientific; 1:400) was incubated for 1 h at room temperature. Vectashield Antifade Mounting Medium with DAPI (4′6-diamidino-2-phenylindole, Cat.no. H-1200, Vector Laboratories) was used to mount coverslips and counterstain nuclei. Sections were viewed and photographed with an Eclipse 90i microscope equipped with a Ds-Ri1 Digital Microscope Camera (Nikon Instruments Inc.). Z-stacking was used to improve the depth of the field of digital images. All images for a given antibody were collected using identical settings for each tissue section.

### 2.5. RNA-seq Analysis and Quantitative Real-Time RT-PCR (RT-qPCR)

Total RNA was isolated from normal skin pericytes, burn eschar pericytes, and normal skin fibroblasts using RNeasy Mini Kit (Qiagen Inc, Valencia, CA, USA) following the manufacturer’s instructions. The quantity and quality of RNA were determined by measuring the OD 260/280 ratio using an ND-100 spectrophotometer (Nanodrop Technologies Inc., Wilmington, DE, USA) and by capillary electrophoresis using an Agilent 2100 BioAnalyzer (Santa Clara, CA, USA). The RNA with a RIN value >7 was used for RNA-seq analysis and RT-qPCR assays. RNA-seq analysis was conducted in the Genomics, Epigenomics, and Sequencing Core at the University of Cincinnati. The polyA RNA was isolated from ~200 ng total RNA using NEBNext Poly(A) mRNA Magnetic Isolation Module (New England BioLabs, Ipswich, MA, USA) combined with SMARTer Apollo NGS library prep system (Takara, Mountain View, CA, USA) for automated polyA RNA isolation. The NEBNext Ultra II Directional RNA Library Prep Kit (New England BioLabs) was used for library preparation under the PCR cycle number of 9. After library QC analysis and qPCR quantification, individually indexed and compatible libraries were proportionally pooled and sequenced using HiSeq platform (San Diego, CA, USA). Under the sequencing setting of single read 1 × 51 bp, about 25 million pass filter reads per sample were generated. Sequence reads were demultiplexed and exported to fastq files using CASAVA 1.8 software (Illumina). The sequencing data were submitted to the Sequence Read Archive (SRA) of National Center for Biotechnology Information (NCBI), and are accessible through the GEO accession number; GSE140926.

To analyze differential gene expression, sequence reads were aligned to the reference genome using the TopHat aligner, and reads aligning to each known transcript were counted using Bioconductor packages for next-generation sequencing data analysis. Sequence reads were demultiplexed and exported to fastq files using CASAVA 1.8 software (Illumina). Cuffdiff or DESeq2 was the tool used for differential expression. The differential expression analysis between different sample types was performed using the negative binomial statistical model of read counts as implemented in the edgeR Bioconductor software package. The cluster analysis of all genes differentially expressed in individual comparisons is performed using the Bayesian infinite mixture models, which was analyzed by The Laboratory for Statistical Genomics and Systems Biology at the University of Cincinnati.

### 2.6. Quantitative Real-Time RT-PCR (RT-qPCR)

Total RNA isolated (RNeasy Mini Kit, Qiagen Inc.) from normal skin pericytes, burn eschar pericytes, and dermal fibroblasts was subjected to RT-qPCR to analyze the expression levels of the target genes. Up to 400 ng of total RNA was used for cDNA synthesis using random primers (100 ng; Invitrogen) and a M-MLV reverse transcriptase (Invitrogen). All gene-specific probes were FAM Taqman MGB probes and used a Taqman Universal PCR Master Mix. Taqman probes for the following human gene products were purchased from Thermo Fisher: *FOXE1* (Hs00916085_s1), *POSTN* (HS01566750_m1), *ADAM12* (HS01106101_m1), *TGF-β1* (HS00998133_m1), *CD248* (endosialin; HS00535586_s1), *FN1* (HS01549976_m1), and *GAPDH* (Hs02786624-G1). GAPDH was used as a housekeeping gene control for all experiments. Real-time qPCR was performed using a StepOnePlus Real-Time PCR System using the following protocol: enzyme activation at 95 °C for 10 min, then 40 cycles of PCR at 95 °C for 15 s and 60 °C for 1 min. The comparative critical cycle (Ct) method was used to determine the expression levels of target genes after normalization to GAPDH expression. The data are presented as fold change pericyte expression levels compared to fibroblast expression ± SEM.

### 2.7. Cell Proliferation (MTT) Assay

Primary normal and burn eschar skin-derived pericytes (5 × 10^4^) seeded on a 24 well plate were grown overnight in pericyte media (Zenbio). The following day, the cells were washed with 1× PBS and switched to DMEM media (Thermo Fisher) containing 0.1% dialyzed FBS for 24 h. Cells were then treated with or without TGF-β1 (PeproTech, Rocky Hill, NJ, USA) at a concentration of 10 ng/mL in 0.1% dialyzed FBS DMEM for another 24 h. After this treatment period, the MTT assay was performed using the CellTiter 96^®^ Non-Radioactive Cell Proliferation Assay (Promega Corporation, Madison, WI, USA). Cells were placed in fresh media and dye solution according to the manufacturer and incubated at 37 °C for two hours, followed by the addition of solubilization/stop solution and another incubation for 1.5 h at 37 °C. Results were obtained using a SpectraMax 384 Plus plate reader (Molecular Devices, San Jose, CA, USA) at 570 nm of 200 μL aliquots placed into a Falcon Tissue Culture Treated 96 well plate. Optical densities were obtained using SoftMax Pro v3.1.2 software. Data are represented as fold change in proliferation ± SEM.

### 2.8. Enzyme-Linked Immunosorbent Assay (ELISA)

IL-6 and IL-8 human ELISA kits were purchased from Thermo Fisher Scientific and were run according to the manufacturer’s instructions. Briefly, supernatant from pericytes grown in 6 well plates was collected in triplicate. Then, 50 μL of supernatant was added to each well along with 50 μL of biotinylated antibody reagent. The plate was covered and incubated at room temperature for two hours. Following washes, 100 μL of a streptavidin-HRP solution was added to each well and incubated at room temperature for 30 min. Following additional washes, 100 μL of TMB substrate solution was added to each well, incubated in the dark for 30 min, and the reaction was terminated with the addition of 100 μL stop solution. The plate was read at 450 nm using a SpectraMax 384 plate reader. Data are presented as an average concentration ± SEM. 

### 2.9. In Vitro Wound Healing Assay

Primary normal and burn eschar skin-derived pericytes were cultured to confluence in Falcon Tissue Culture Treated six-well plates and allowed to grow for 24 h at 37 °C. On each plate, a horizontal line was drawn across each well to image the same area consistently. The cells were then washed with 1× PBS and the media was changed to DMEM containing 0.1% dialyzed FBS and incubated for 24 h at 37 °C. Following this incubation, media was removed and using a P1000 pipet tip, a vertical wound was made across the monolayer of cells. Each well was then washed twice with 1× PBS and imaged for time zero. Cells were then changed to DMEM with 0.1% dialyzed FBS for 24 h at 37 °C. After this treatment period, the media was collected for future use, and the cells were washed twice with 1× PBS. A 24-h treatment image was taken. Using ImageJ software (NIH), the distance of the wounds was measured for both time zero and 24 h post-treatment. These distances were used to calculate % closure as such: (Time zero distance − 24-h distance/Time zero) distance × 100. Cell migration was expressed as a fold change compared to no treatment cells ± SEM.

### 2.10. Cell Contraction Assay

Primary normal and burn eschar pericytes along with fibroblasts were grown in cell culture flasks to a confluent monolayer, quiesced for 24 h in DMEM with 0.1% dialyzed FBS, trypsinized, and collected at a concentration of 2 × 10^6^ cells/mL. The cells were mixed at a ratio of 2:8 with a collagen solution (Cell Bio Labs Inc. San Diego, CA, USA) and incubated for 1 h at 37 °C with 5% CO_2_. After incubation, fresh DMEM with 0.1% dialyzed FBS was added on top of the collagen gels, and images were taken at time zero and again 24 h later. Cell contraction was measured using ImageJ-NIH software analysis, and data are presented as fold change contraction as compared to day 0 images ± SEM.

### 2.11. Statistical Analysis

Statistical analysis of the RNA-seq data was performed by The Laboratory for Statistical Genomics and Systems Biology at the University of Cincinnati. RNA-seq results were considered significant with a false discovery rate (FDR) <0.1. All the other data are presented as mean ± standard error of the mean. Student’s t-test and One-way analysis of variance (ANOVA) were used and *p* < 0.05 was considered statistically significant.

## 3. Results

### 3.1. Isolation of Pericytes from Burn Eschar

Pericytes play an important role in the process of wound healing. Here we seek to isolate and characterize pericytes in the eschar of burn wounds. First, we characterized the morphology and phenotype of pericytes isolated from burn wounds. Pericytes were isolated from normal skin, or the burn wound eschar by flow cytometry using the following cell marker profiles; CD73+, CD105+ CD146+, CD34−, CD45−, and CD56−. After incubation in pericytes growth media the cells were analyzed by flow cytometry for the pericyte marker CD146 and NG2 and stem cell markers CD105 and CD73. The cells were also stained for the endothelial marker CD34 and skeletal muscle marker CD56. The majority of isolated cells express the pericytes markers CD146 (95%), CD73 (95%), CD105 (55%), and NG2 (23%) but not skeletal muscle (CD56) or endothelial (CD34) cells markers both less than 5% (Figure 1a) suggesting the isolated cells are pericytes. A study by van der Veen et al. isolated stem cells from burn eschar [39]. In this study, the stem cells isolated were positive for CD90, CD105 and CD73 and negative for CD31, CD45, CD14, and CD79a which have a similar CD expression profile to the cells that were isolated in the present study. Phase contrast images of pericytes isolated from both normal skin and burn eschar tissues display a cell body with a prominent nucleus containing small amount of cytoplasm with several long processes (Figure 1b). To further verify that the isolated cells are pericytes, the cells were stained for pericyte markers CD146 and, NG2, as well as fibroblast marker S100A4 (Figure 2). Here we show that greater than 90% of the pericytes expressed CD146 and nearly all the pericytes stained for NG2 (Figure 2a–c). To verify whether fibroblast may have been isolated along with the pericytes the cells were analyzed for the expression of the fibroblast marker S100A4. High expression of S110A4 is seen on fibroblast (Figure 2a,d). Pericytes isolated from normal skin and burn eschar, showed only a few cells with a very low level of S100A4 staining (Figure 2d). When the pericytes isolated from the burn eschar were compared to normal skin pericytes, there was a slight increase in the number of S100A4 positive cells, but this increase was not significant (Figure 2d). Compared to fibroblasts the S100A4 staining in the pericytes was only 11.67% ± 3.85; normal skin and 14.49% ± 4.51; burn eschar (Figure 2d). Interestingly, pericytes isolated from burned eschar but not from normal skin showed expression of some smooth muscle markers such as SM22 but not MHY11 or Calponin-1 (Figure 2e). Compared to normal skin pericytes with no SM22 staining, pericytes isolated from burn eschar showed SM22 staining up to 60%. Meanwhile, another smooth muscle marker MHY11 or Calponin-1 was absent from pericytes isolated from either normal skin or burn eschar suggesting a plasticity of fate in burn eschar pericytes. Taken together the data indicates that the significant number of cells isolated are pericytes and these cells should have a MSC phenotype since they express the MSC markers CD90, CD105, and CD73.

### 3.2. Burn Eschar Pericytes Overexpress Inflammatory Cytokines

Next, we determine whether the burn environment affected pericytes gene expression. Using RNA-seq, we identified 443 differentially expressed genes between normal- and burn eschar-derived pericytes. While comparing normal skin pericytes to normal skin fibroblasts, we identified 1021 distinct genes, however comparing burn eschar pericytes to normal skin fibroblasts we identified 2449 differential genes (Appendix A). From these sets of genes, we identified six pro-inflammatory genes and two anti-inflammatory genes to be upregulated (Table 2) specifically in burn eschar pericytes. Using an ELISA assay, we analyzed the protein expression of these genes in normal skin and burn eschar derived pericytes. We found that both interleukin-6 (IL-6; 5 fold) and interleukin-8 IL-8; 3.75 fold) protein levels were significantly higher in burn eschar pericytes than normal skin pericytes (Figure 3). Utilizing real-time RT-PCR analysis, we found that TGF-β1 gene expression was significantly higher (38%) in the burn eschar derived pericytes than the normal skin-derived pericytes (Figure 4a). Rustenhoven et al. showed that TGF-β1 treated pericytes induced the secretion of IL-6 [40]. In addition, they showed that pericyte stimulation with TGF-β1 and IL-1β has a synergistic release of IL-6 [40] which correlates with our finding that IL-1βis upregulated (Table 2) and a 5 fold increase in IL-6 expression (Figure 3a). MyD88, an upstream regulator of interleukin-1 (IL-1) and tumor necrosis factor-α (TNF-α) expression [41,42] was also examined. We found expression of MyD88 was increased in the burn eschar derived-pericytes compared to normal skin derived-pericytes (Figure 4b). Although the increase did not reach significance, the results are suggestive that the burn eschar derived-pericytes increase pro-inflammatory cytokines in part through the activation of MyD88. A disintegrin and metalloprotease 12 (ADAM12) has been found to be upregulated in chronic ulcers and may contribute to a decrease in keratinocyte migration and proliferation [43,44]. Furthermore, ADAM12+ cells were found to be progenitors of collagen-overproducing cells [45]. When ADAM12 gene expression was examined, we found that expression was significantly greater (175%) in the burn eschar derived pericytes than the normal skin-derived pericytes (Figure 4c). These data suggest that the burn eschar stimulates pericyte secretion of the pro-inflammatory factors TGF-β1 and IL-1β. These factors further promote an enhanced secretion of pro-inflammatory cytokine IL-6. Taken together, burn eschar derived-pericytes have a pro-inflammatory phenotype. The overexpression of ADAM12 is suggestive of the possibility for pericyte differentiation into a pro-fibrotic phenotype.

### 3.3. Burn Eschar Derived-Pericyte Express FOXE1

The expression of forkhead box E1 (FOXE1) has been primarily found in thyroid cells and is essential for thyroid development [46,47]. In our RNA-seq analysis, we found that FOXE1 expression was highly upregulated in both normal skin (601.95 fold increase) and burn eschar pericytes (1640.93 fold increase) when compared to fibroblasts (Appendix A). To verify the expression of FOXE1 in pericytes, real-time RT-PCR was performed. Here we show that FOXE1 expression in pericytes isolated from normal skin was minimal, but the expression in the pericytes from burn eschar was significantly higher (Figure 5a). There was a 12 fold increase in FOXE1 transcription in the burn eschar derived pericytes compared to normal skin pericytes (Figure 5a). Immunocytochemistry was performed on the cells to verify that this increase in mRNA expression translated to an increase in protein expression (Figure 5b). Quantification of cells expressing FOXE1 between normal skin pericytes and burn eschar pericytes showed that FOXE1 protein expression is significantly increased in burn eschar pericytes (Figure 5c). This is the first observation of FOXE1 expression in pericytes. The function of FOXE1 outside of the thyroid is not well understood. Since FOXE1 is involved in thyroid development and differentiation, it may play a role in regulating the differentiation state of pericytes during the wound healing process which will be investigated in future studies.

### 3.4. Burn Eschar Derived-Pericyte Express Fibroblast Markers

Studies have shown that pericytes play a role in fibrosis by differentiating into a fibroblast-myofibroblast like phenotype. The signaling mechanisms that promote pericyte differentiation to a fibroblast-myofibroblast phenotype are not well understood. In RNAseq data, we found that fibronectin, endosialin, and periostin gene expression is overexpressed in burn eschar derived-pericytes (Appendix A). Using real-time RT-PCR, we found that the burn eschar derived-pericytes had a significant increase in fibronectin (FN1) expression (60%) compared to normal skin derived-pericytes (Figure 6a). Endosialin (CD248) is dynamically expressed on pericytes and fibroblasts during tissue development and inflammation. It has been shown that tissue with high endosialin mRNA expression has the protein present, whereas low mRNA expression fails to translate to protein detection [48]. A study by Smith et al. [49] found that endosialin knockout mice were protected from fibrosis due to less pericyte differentiation towards a myofibroblast phenotype. Here we show that pericytes isolated from burn eschar have a significantly (0.82 fold) higher expression of endosialin (CD248) mRNA than normal skin pericytes (Figure 6b). Periostin is a matricellular protein that has been found to play a role in dermal fibroblast proliferation and differentiation into myofibroblast [50,51,52]. Here we found that periostin gene expression was significantly higher (0.75 fold) in the burn eschar derived pericytes than normal skin pericytes (Figure 6c). When mRNA expression of periostin and fibronectin was compared between pericytes isolated from burn eschar and fibroblasts, we found the pericytes had a 15 fold increase in periostin (Figure 6d) and 3 fold increase in fibronectin 1 (Figure 6e). Next, periostin protein expression in vivo was determined. Healthy normal human skin or burn eschar were processed for histological analyses of pericytes and periostin. Tissues were incubated with CD146 and periostin antibody then visualized using fluorescent secondary antibodies. We show in normal skin, pericytes (green) had very little periostin (red) staining (Figure 6f). In the burn eschar, significant staining of periostin (red) was observed throughout the tissue (Figure 6f). The pericytes (green) in the burn eschar tissue had a high expression of periostin compared to normal skin pericytes (Figure 6f). Taken together these data are suggestive that the burn wound environment promotes the expression of genes and proteins in pericytes that are known mediators of fibroblast to myofibroblast transition.

### 3.5. Burn Eschar Derived-Pericytes Have an Increased Contractile Response

Here we determined whether the burn eschar isolated pericytes have a fibroblast-like phenotype by studying the functional properties of proliferation, migration, and contraction. Using the MTT assay we did not observe a difference in proliferation between fibroblast, pericytes isolated from normal skin or pericytes from burn eschar (Figure 7a). Using the scratch assay to analyze migration, we did not see a significant difference between fibroblasts and pericytes (Figure 7b). Since collagen contraction is one of the major functions of fibroblasts and myofibroblasts, we determined the ability of the cells to contract a collagen lattice. Here we show, pericytes isolated from the skin or burn eschar both had an increased ability to contract collagen compared to fibroblasts (Figure 8a). We show that the pericytes from normal skin had a 5 fold and pericytes from burn eschar had a 6 fold change in contraction compared to fibroblasts (Figure 8b). Taken together these data suggests that burn eschar pericytes possess the ability to contract the wound matrix.

## 4. Discussion

The processes involved in wound healing that leads to normal tissue repair are well identified, but our understanding of how dysregulation of inflammatory and reparative cells leads to fibrosis is not well understood. The inflammatory process is required to promote wound healing, but an extended and persistent pro-inflammatory cascade leads to fibrotic scarring. Traditionally, the fibroblast and myofibroblast have been presumed to be the principal cell mediating fibrosis [53]. During the granulation phase of wound healing, local fibroblasts begin migrating from the periphery into the provisional scar tissue. As they move into the provisional matrix, they begin to differentiate into a cell phenotype known as myofibroblast, which are responsible for wound contraction. Upon closure of the wound, the myofibroblast undergoes apoptosis [54]. On the other hand, the persistence of myofibroblasts during the remodeling phase promotes hypertrophic scarring and leads to the development of fibrosis. Recent studies have indicated that other cells may be involved in fibrotic disease [55,56,57]. Pericytes are cells that have been found to possess stem cell-like properties with a capacity to differentiate into cell-states with pro-regenerative properties in addition to their essential role in regulating angiogenesis [20,25].

Pericytes are associated with the microvasculature and facilitate vascular stabilization, maintain vessel integrity and regulate vascular tone. During angiogenesis activated pericytes detach from the vessel to facilitate endothelial dissociation and initiation of angiogenesis then reattach to promote vessel stabilization and maturation. During would healing pericytes have been found to detach from microvessels [11,58]. Pericyte detachment and subsequent activation is thought to promote dedifferentiation into a stem cell-like phenotype. In this state, pericytes have the potential to differentiate into various cells depending on ques from neighboring cells or interaction with specific signaling molecules [20,23,59]. The pluripotency of pericytes have made them a novel candidate for tissue repair. Pericytes implantation has been successful in repairing various tissues including muscle, heart, and brain [16,29,60,61]. Pericytes have also been implicated in promoting fibrosis in various tissues [36,37,55,62,63]. Studies by Birbrair et al. suggest two populations of pericytes type 1 and type 2 [64]. Type-1 pericytes when exposed to TGF-β are fibrogenic [35]. Type-2 pericytes were found to promote angiogenesis and tissue repair [60,65]. Understanding the signaling pathways that promotes pericytes to a collagen producing phenotype, myofibroblast/fibroblast, will enable the modulation of pericytes into a more regenerative phenotype than a fibrotic-promoting state.

Fibrosis is an excessive amount of extracellular matrix (ECM) that accumulates in the wound tissue and leads to the loss of organ function. Fibrosis is viewed as excessive scar formation. During the granulation phase of wound healing, myofibroblasts initially deposit fibronectin and collagen III, forming the provisional matrix. Late in the granulation phase and the remodeling phase of wound healing, fibroblasts replace the provisional ECM with collagen I and elastin [66]. Fibrosis occurs when there is excessive production of ECM by myofibroblasts, which compromises normal tissue function, eventually leading to cell injury and death, which can further trigger myofibroblast activation continuing the fibrotic response cascade. Biologically active TGFβ-1 is one of the major cytokines needed to generate α-SMA-positive myofibroblasts. Pathologies of over scarring or fibrosis include hypertrophic scars and keloids [67]. Recent studies have implicated pericytes in contributing to the fibrotic response [68,69]. Understanding the process and cells involved in fibrosis will allow for the identification of critical signaling pathways as potential therapeutic targets to circumvent fibrosis and promote normal tissue repair.

In the U.S., acute thermal injuries occur in over 2 million people a year. Of these, approximately 75,000 are hospitalized and about 14,000 die due to burn injury. World-wide an estimated 180,000 deaths a year are caused by burns [70]. Thermal burns over 20% of total body surface area results in burn shock which occurs in the first 18 hrs then there is a prolonged period of hypermetabolism and chronic inflammation which creates a high level of circulating cytokines [71]. Monocytes and neutrophils along with endothelial cells secrete the inflammatory cytokines TGF-β1, TNF-α, IL-1, IL-6, IL-8, and MCP-1 after burn injury. Excessive production of these cytokines can lead to systemic inflammatory response syndrome. In the pericytes isolated from burn eschar, a significant increase in TGF-β1, IL-1, and IL-6 was observed. This is highly suggestive that the wound environment is promoting pericytes secretion of inflammatory cytokines enhancing the inflammatory response. The mechanism promoting pericytes secretion of inflammatory cytokines will be useful in deducing the factors involved in excessive fibroplasia.

The main goals of this study were to characterize resident pericytes in burn eschar and determine whether these pericytes have distinct, identifiable characteristics. We have isolated pericytes from burn eschar (Figure 1 and Figure 2) without enzymatic digestion and have shown differences exist between these cells and normal skin pericytes. The ability to isolate pericytes from wound eschar is of significant importance as it provides an avenue to characterize pericytes from different types of wounds and provides an opportunity to study the role and function of pericytes in different wound healing conditions. We show pericytes isolated from burn eschar have an increased expression of the pro-inflammatory cytokines IL-6 and IL-8 (Figure 3), TGF-β1 (Figure 4), and IL-1 (Table 2). It is well established that TGF-β1 is the main driver of contractile genes that promotes precursor cells to a myofibroblast phenotype [53]. TGF-β1 induces p38 mediated activation of PI-3K-AKT pathway which confers apoptosis resistance [72]. It can activate the sonic Hedgehog pathway promoting transdifferentiation of hepatic stellate cells into myofibroblasts [73]. IL-1β has been found to increase the expression of TGF-β1 [74] and transform dermal microvascular endothelial cells into myofibroblasts [75]. Taken together, this suggesting that pericyte secretion of TGF-β1 and IL-1βcould promote differentiation of precursor cell to a myofibroblast phenotype. MyD88, an upstream regulator of IL-1 and TNF-α has been shown to impair wound healing by upregulating the expression of pro-inflammatory cytokines and has been shown to contribute to fibrogenesis [76]. MyD88 is also upregulated in pericytes isolated from burn eschar (Figure 4). The activation of Toll-like receptor (TLR)/MyD88 signaling pathway is required for myofibroblast differentiation [76]. It is thought that MyD88 transduce profibrotic signaling through TGF-β receptor transactivation. Here we show that TGF-β1 is upregulated in burn eschar pericytes (Figure 4). Thus, the expression of TGF-β1 and IL-1 by pericytes can promote an autocrine enhancement of inflammatory pathways and potentially exacerbating fibrosis. It could be postulated that pericyte secretion of TGF-β1 could promote an increased number of myofibroblasts found in the wound. Further the possibility exists that high levels of TGF-β1 may contribute to the differentiation of pericytes into myofibroblasts which has been observed in kidney fibrosis [77]. The role TGF-β1 has on pericyte function in wound healing will be the focus of future studies.

We also show for the first time that pericytes from burn eschar express high levels of FOXE1, forkhead box E1 (Figure 5). This finding is of great interest since FOXE1, also known as thyroid transcription factor 2 (Ttf2), is a thyroid-specific transcription factor that belongs to the forkhead/winged-helix family that is essential for thyroid development [78]. The FOX proteins are a superfamily of conserved transcriptional regulators having a highly conserved forkhead box or winged helix DNA binding domain [79,80]. FOXE1 interacts with nucleosomes through its winged-helix DNA binding domain. In the thyroid, FOXE1 is essential during thyroid development and differentiation [81]. FOXE1 binds to promoters of 2 thyroid-specific genes (thyroglobulin and hydroperoxidase) and can act to promote and repress transcription of both genes [82,83,84]. A study by Fernández et al. identified 54 novel FOXE1 target genes, which include ADAMTS 9 and S100A4 [85]. A disintegrin and metalloproteinase with thrombospondin motifs (ADAMTS) 9 is involved in the binding of fibronectin and disrupting fibronectin fibrils [86]. S100A4 is involved in cell cycle progression and differentiation [87]. Here we show that S100A4 is highly expressed in fibroblast and virtually absent in normal skin pericytes (Figure 2). Although S100A4 expression in pericytes isolated from burn eschar is very low compared to fibroblasts, its expression is 25% higher compared to normal skin pericytes (Figure 2d). Thus, this increase in expression could contribute to modifying pericyte phenotype in burn wounds. These findings may suggest that the expression of FOXE1 in the burn eschar pericytes could be involved in S100A4 expression in pericytes isolated from burn eschar. Although we did not observe an upregulation of ADAMTS 9, we did observe a significant increase in ADAMT 12 mRNA expression (Figure 4c). It is not known if ADAMT 12 is a FOXE1 target gene, but overexpression of ADAMT 12 inhibits keratinocyte migration and proliferation causing impairment in wound healing [44]. Thus, overexpression of ADAMTS12 can contribute to the formation of chronic ulcers.

Phenotypically, pericytes from the burn eschar are found to express the fibroblast markers endosialin (CD248), periostin, and fibronectin 1 (Figure 6), which are known mediators of fibroblast to myofibroblast transition. Periostin is a matricellular protein found mainly in collagen-rich tissues, and expression is primarily associated with fibroblasts [50,88] and has been found to play a role in fibroblast proliferation and their differentiation into myofibroblast during wound repair [51,52]. During skin development, periostin is produced and is strongly related to pathological skin remodeling [89]. Yamaguchi et al. showed that serum levels of periostin correlated with fibroblast infiltration in patients with systemic sclerosis [90]. Recently, macrophage/monocyte infiltration has shown to be facilitated by periostin [69], and it can regulate keratinocyte differentiation and proliferation [91]. Periostin is a ligand for integrin αv β3, which promotes activation of FAK/PI3K/Akt pathway [92]. In renal disease, periostin mediates the phosphorylation of FAK, p38, and Erk [93]. During cutaneous wound healing, periostin protein expression is observed at day 3 and peaks between days 7 to 10 [51,52]. Thus, periostin may play a role in ECM remodeling and regulation of inflammation during wound healing. Here we show that periostin mRNA expression in pericytes from burn eschar is two-fold higher than normal skin pericytes (Figure 6c) and fifteen-fold higher than fibroblasts (Figure 6d). These data may suggest that upregulation of periostin in pericytes could modulate pericytes to a myofibroblast phenotype.

To our knowledge, this is the first study to isolate and characterize pericytes from burn wounds. Our findings from this study suggest that the wound environment plays a critical role in regulating the function and phenotype of pericytes during the healing process. Our data suggest that pericytes isolated from burn eschar differentiate into a fibroblast/myofibroblast phenotype due to the high expression of MyD88, fibronectin, periostin, and endosialin. Thus, these pericytes may contribute to fibrosis. In previous studies, including ours, have demonstrated that pericytes play a role in enhancing the wound healing process [28,94,95,96]. The difference between these previous studies and this study is the wound environment. These previous studies looked at pericyte function in excision wounds, which do not incur a significant inflammatory response nor significant tissue damage. This study examined pericytes from acute burn wounds incurring severe tissue damage with increased inflammatory response. Understanding the factors and signaling pathways contributing to the differentiation of pericytes into a fibroblast-like cell may provide a novel treatment to significantly reduce or eliminate fibrosis in burn scars.

This study is also the first to identify the expression of FOXE1 in pericytes isolated from normal skin and burn eschar. This is of great interest since FOXE1 is a regulator of gene expression and essential for thyroid development and differentiation. FOXE1 is predominantly express in the thyroid but expression has also be found in colon, lung, and orofacial tissue [97,98,99]. FOXE1 has not been observed to play a role in fibrosis but a number of FOX member have shown to be involved in fibrosis. FOXP1has been shown to enhance the Wnt/β-catenin signaling pathway promoting the expression if collagen, α-SMA and fibronectin [100]. In heart, cardiac fibroblasts differentiation into myofibroblast in a FOXO1 dependent manner [101]. Further, authors show that the TGF-β induced the activation of FOXO1. In skin, knockdown of FOXO1 enhanced wound healing through enhanced keratinocyte migration, reduced inflammatory response and decreased collagen density [102]. FOXE1 may play a key role in pericyte differentiation into various phenotypes or the secretion of collagen. Very little is known regarding the signaling pathways involved in the differentiation of pericytes into other cell lineages. Thus, activation of FOXE1 may be a contributing factor to pericyte-myofibroblast differentiation. Understanding the expression and function of FOXE1 in pericytes will further our understanding of the signaling pathways regulating pericytes function. Further studies will be performed to analyze FOXE1 function and its implication on pericyte differentiation.

## 5. Conclusions

We report here for the first time that pericytes can be isolated from discarded normal skin and burn eschar tissues without the utility of the enzyme digestion process. We show that pure population of pericytes can be obtained, which can be utilized for biological analyses. Our studies show significant differences in the gene expression between normal skin pericytes and burn pericytes. We also observed a significant difference in gene expression patterns between fibroblasts and pericytes leading to the identification of the gene FOXE-1. FOXE-1 expression was noted specifically in pericytes, which may serve as a new marker for identifying skin pericytes. We also provide evidence that pericytes isolated from burn eschar express high levels of the inflammatory cytokines namely, TGF-β1, IL-6, and IL-8. It is well established that these cytokines/chemokines play a significant role in fibrosis. In addition, previous studies, have demonstrated that pericytes have the ability to express collagen and differentiate into a myofibroblast phenotype. Our results show that pericytes from burn wounds, express higher levels of pro-inflammatory cytokines, and may be a source of myofibroblasts contributing to burn scar contractures.

## Figures and Tables

**Figure 1 jcm-09-00606-f001:**
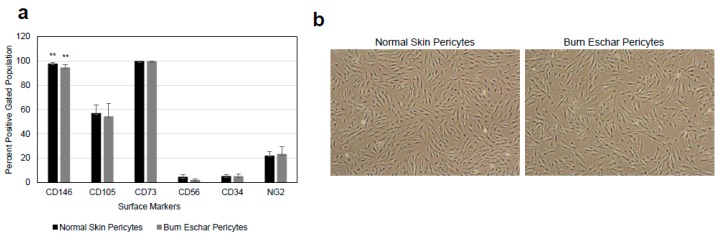
(**a**) Isolation and characterization of pericytes from skin. Cells grown in pericyte growth media were analyzed for the pericytes marker CD146 and neuron-glial 2 (NG2) and stem cell markers CD105 and CD73. The cells were also stained for the endothelial marker CD34 and skeletal muscle marker CD56. These cells did not express the endothelial marker CD34 or smooth muscle marker CD56 indicating that the isolated cell population are of pericyte lineage. Pericytes isolated from each of six different patients undergoing debridement after burn injury and breast reduction surgeries were used for flow analysis. Statistical analyses were performed using one-way analysis of variance (ANOVA) and *p* < 0.05 was considered statistically significant. ** *p* < 0.01. (**b**) Shown are representative phase-contrast images of pericytes isolated from normal skin and burn eschar tissues.

**Figure 2 jcm-09-00606-f002:**
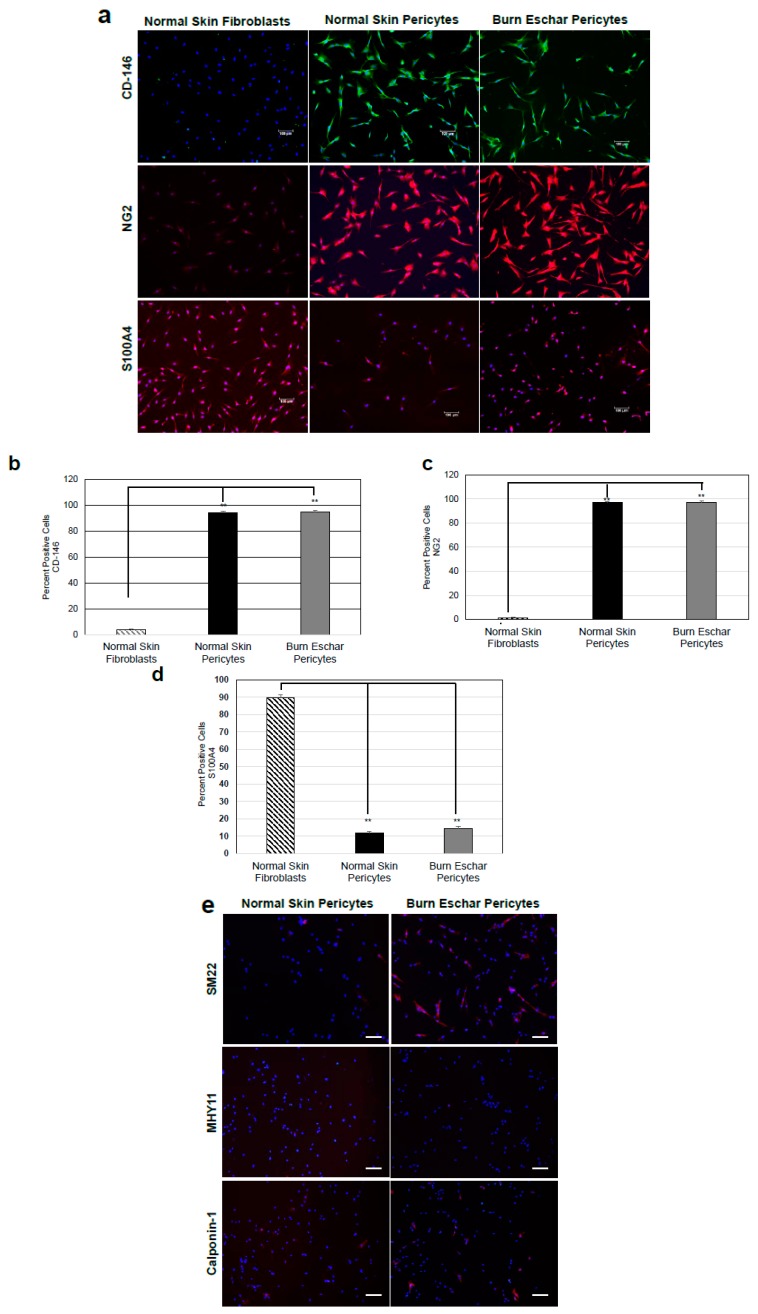
Isolated cells are a heterogeneous population of pericytes. (**a**) Cultured pericytes from normal skin and burn eschar were analyzed for expression of the pericyte markers CD146 and NG2 along with the fibroblast marker S100A4. Representative images of the experiments performed on three different cultures derived from three different patients for normal skin pericytes, burn eschar pericytes and fibroblasts are shown. Images were captured using Eclipse 90i microscope and photographed with a DS-Qi1MC Digital Microscope. Scale bar: 100 μM. Quantitative analysis of (**b**) CD146, (**c**) NG2, and (**d**) S100A4 was performed. Quantitative analysis was performed in triplicate at 10x high powered fields using NIS-Elements AR3.1 software. Data represented as mean ± SEM of two independent studies. Statistical analysis was performed using Student’s *t* test with *p* < 0.05 considered statistically significant. ** *p* < 0.01. The results show pericytes are greater than 90% positive for CD146 and NG2 and less than 15% positive for S100A4. The fibroblasts are greater than 90% positive for S100A4 and less than 10% positive for CD146 and NG2. (**e**) Shown is the representative images on the staining performed on normal skin pericytes and burn eschar pericytes derived from two different patients using SM22, myosin smooth muscle heavy chain (MHY)11, and Calponin-1 are shown. Scale bar: 30 µM.

**Figure 3 jcm-09-00606-f003:**
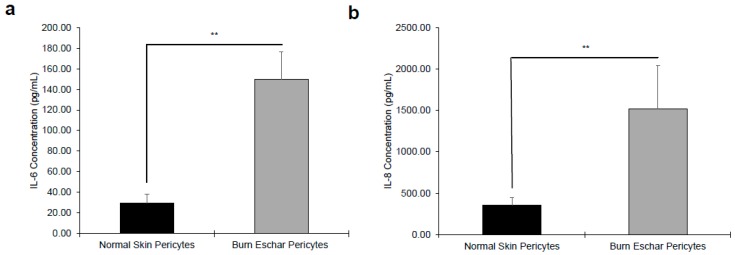
Burn Eschar pericytes express high levels of pro-inflammatory cytokines. ELISA analysis of (**a**) IL-6 and (**b**) IL-8 expression by normal and burn eschar pericytes. ELISA analysis was performed using normal skin pericytes derived from three different patients and burn eschar pericytes derived from four different patient samples. The results indicate that pericytes isolated from the burn eschar express significant higher amounts of the pro-inflammatory cytokines IL-6 and IL-8 compared to normal pericytes. Statistical analysis was performed using ANOVA with *p* < 0.05 considered statistically significant. Data is represented as mean ± SEM values. ** *p* < 0.01. Interleukin- 1alpha (IL-1α); Tumor necrosis factor alpha-induced protein 3 (TNFαIP3); chemokine ligand- 14 (CXCL-14); nuclear factor of kappa light polypeptide gene enhancer in B cells inhibitor, zeta (NFκBIZ); nuclear factor of kappa light polypeptide gene enhancer in B cells inhibitor, alpha (NFκBIα).

**Figure 4 jcm-09-00606-f004:**
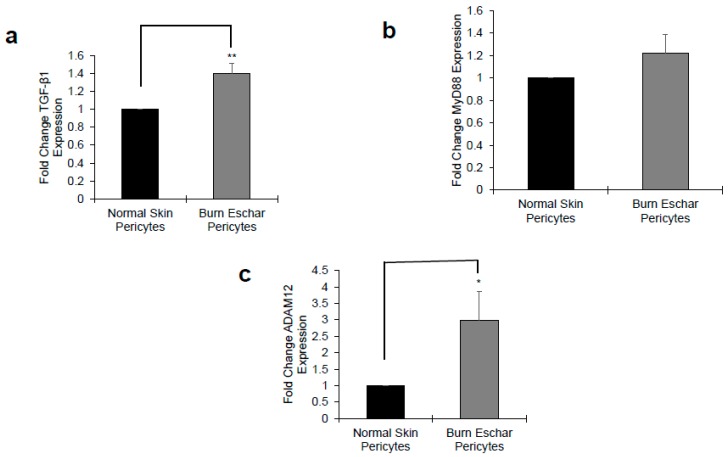
Burn eschar pericytes express high levels of pro-fibrotic genes. Normal and burn eschar pericytes were analyzed for transcription of (**a**) transforming growth factor (TGF)-β1, (**b**) myeloid differentiation factor 88 (MyD88), and (**c**) A disintegrin and metalloprotease 12 (ADAM12) using real time RT-PCR. Burn eschar pericytes display significantly higher expression of TGF-β1 and ADAM12 than normal pericytes. Statistical analysis was performed using ANOVA with *p* < 0.05 considered statistically significant. Values are means ± SEM of three independent studies using three different cultures for both the cell types performed in triplicate. * *p* < 0.05; ** *p* < 0.01.

**Figure 5 jcm-09-00606-f005:**
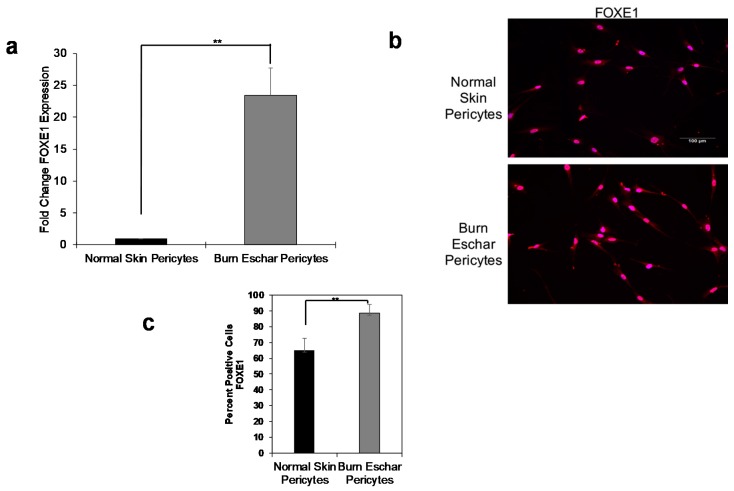
Pericytes express forkhead box E1 (FOXE1). (**a**) mRNA expression of FOXE1, (**b**) immunofluorescence of FOXE1 protein expression, and (**c**) Quantitative analysis of FOXE-1 protein expression. Both normal and burn eschar pericytes express FOXE1 but the burn eschar pericytes have a significantly greater expression of FOXE1 than normal pericytes. Three different cultures of pericytes derived from three different patients of normal skin and burn eschar tissues were utilized for real time RT-PCR; two different cultures of normal skin and burn eschar was used for immunofluorescence. Quantitative analysis was performed on FOXE1 stain in 3 10× high powered fields using NIS-Elements AR3.1 software. Data represented as mean ± SEM of two independent studies. Scale bar: 100 μM. Statistical analysis was performed using Student’s *t* test with *p* < 0.05 considered statistically significant. ** *p* < 0.01.

**Figure 6 jcm-09-00606-f006:**
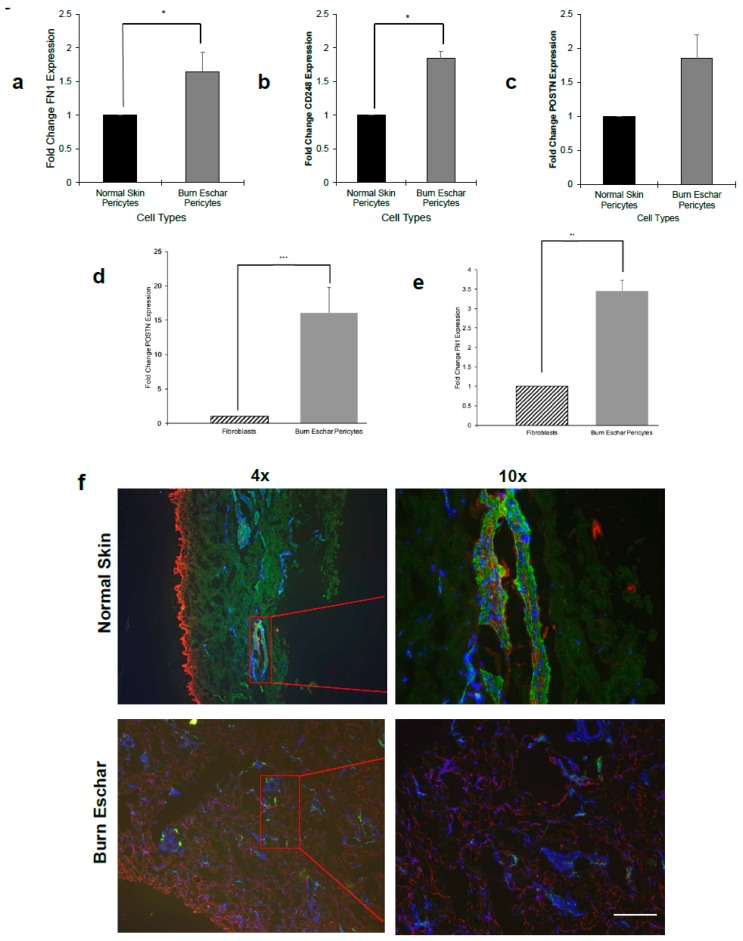
Burn eschar pericytes express fibrotic proteins. mRNA expression of (**a**) fibronectin (FN1), (**b**) endosialin (CD248), and (**c**) periostin (POSTN) between normal skin and burn eschar pericytes was determined using real time RT-PCR. mRNA expression of (**d**) periostin (POSTN) and (**e**) fibronectin (FN1) was compared between fibroblasts and pericytes using real time RT-PCR. (**f**) Immunohistochemistry of normal tissue and burn eschar was performed using antibodies directed towards pericytes (green), periostin (red), and nuclei (blue). Shown here are representative images of three different experiments performed on three different patient derived normal skin and burn eschar tissue sections. Scale bar: 100 µM. These data show that expression of the fibrotic proteins fibronectin, endosialin and periostin is significantly higher in burn eschar pericytes compared to either normal pericytes or fibroblasts. Real time RT-PCR was performed using three cultures derived from normal skin, burn eschar, and normal fibroblasts were used. Data are presented as mean ± SEM values of three independent experiments performed in triplicate. Statistical analysis was performed using ANOVA with *p* < 0.05 considered statistically significant. * *p* < 0.05; ** *p* < 0.01; *** *p* < 0.001.

**Figure 7 jcm-09-00606-f007:**
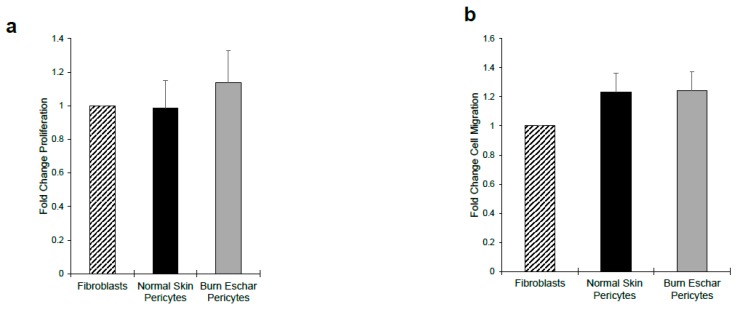
Cell proliferation and migration is unchanged between normal skin and burn eschar pericytes. Analysis of (**a**) cellular proliferation using the MTT assay and (**b**) in-vitro migration using the scratch assay shows no difference between normal and burn eschar pericytes. Cell proliferation assay was performed using five different cultures of normal skin- and burn eschar-derived pericytes and cell migration assay was performed using six different cultures each of normal skin- and burn eschar-derived pericytes. Statistical analysis was performed using one-way ANOVA with *p* < 0.05 considered statistically significant. Data is represented as mean ± SEM values.

**Figure 8 jcm-09-00606-f008:**
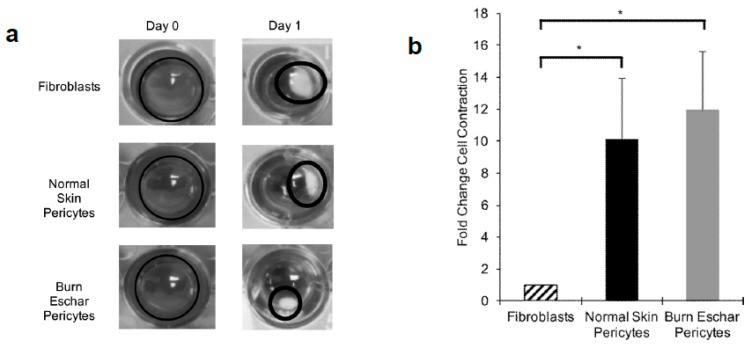
Burn eschar pericytes display increased cellular contractility. (**a**) Collagen contraction assay between fibroblasts and pericytes isolated from normal skin and burn eschar. Cellular contractile ability of normal skin pericytes and burn eschar pericytes was compared to normal skin-derived fibroblasts. Shown here is the representative images of three independent experiments performed in triplicate. (**b**) Quantitative analysis of contracted collagen lattice normalized to the average area of contraction seen in fibroblasts, set as a baseline value of 1. Each data point represents the mean ± SEM of the averages of triplicate reads for each culture derived from the three different patient samples. Statistical significance was determined using one-way ANOVA. * *p* < 0.05. Data indicates that burn eschar pericytes have a greater contractile capability than fibroblast.

**Table 1 jcm-09-00606-t001:** Demographic information on burn eschar and normal skin tissues used for pericytes isolation.

Strain no.	Age	Race	Sex	Type of Burns	Body Site Debridement/Excision/Surgical Procedure
**Burn Eschar**					
Burn 8	11 yrs	Other	Female	Scald	Bilateral Lower Legs
Burn 18	2 months	Black/Non-Hispanic	Female	Contact	Left thigh, Foot and Leg
Burn 20	2 yrs	Hispanic	Male	Scald	Anterior Trunk
Burn 21	16 months	White/Non-Hispanic	Female	Scald	Scalp
Burn 22	9 yrs	White/Non-Hispanic	Male	Female	Left Flank, Left Leg and Left Axilla
Burn 23	9 yrs	White/Non-Hispanic	Male	Female	Chest and Abdomen
**Normal Skin**					
Normal Skin 1	16 yrs	White	Female	N/A	Breast Reduction
Normal Skin 4	14 months	White	Female	N/A	Excess Tissue From Right Thigh Was Used After Using for a Split Skin Thickness Graft
Normal Skin 8	16 yrs	Black/Non-Hispanic	Female	N/A	Breast Reduction
Normal Skin 9	13 yrs	White	Female	N/A	Breast Reduction
Normal Skin 11	16 yrs	African American	Female	N/A	Breast Reduction
Normal Skin 13	18 yrs	Biracial White/Black	Female	N/A	Breast Reduction

**Table 2 jcm-09-00606-t002:** Expression profile of pro- and anti-inflammatory genes in burn eschar pericytes from RNA-seq analysis.

Pro-Inflammatory Genes	Fold Change	Ant-Inflammatory Genes	Fold Change
IL-6	6.74 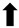	NFκBIZ	2.50 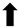
IL-1β	48.39 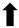	NFκBIα	2.01 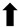
IL-1α	29.52 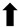		
IL-8 (CXCL8)	15.69 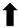		
TNFαIP3	8.38 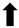		
CXCL-14	0.16 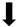

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
