# Peer review of "Characterization of Burn Eschar Pericytes"

_jcm, 2020, doi:10.3390/jcm9020606_

Round 1

Reviewer 1 Report

In this study Authors report that burn eschar pericytes, a subpopulation of mesenchymal stem cells (MSCs), are inflammatory and may contribute to burn scar contractures and fibrosis, representing a population of adult stem cells mobilized in response to the burn injury. Results from these studies provide clues to pericytes in the burn wound environment for better healing outcome in reducing scar contractures, their role in vessel formation and how they can affect the wound healing process.

Strong Point:

1) Previous and even recent published articles, some of them not cited by the authors: Vincent C. van der Veen et al., Cell Transplantation, 2012; Popescu et al., Rom J Morphol Embryol 2011, Mills et al., Cells 2013 had reviewed the literature in this area. The authors should amplify their Bibliography including additional references (see above) to give additional information in the final interpretation of findings for our understanding the relationship of pericytes dissociated from vessels, their hyperactive state in response to the various factors secreted by activated immune cells and their differentiation into myofibroblasts promoting excessive fibroplasia.

2) Authors should amplify their text, discussion and conclusions including information derived from other reported studies, to give additional information in the final interpretation of findings for understanding the the relationship between pericytes from burn wounds, expression levels of pro-inflammatory cytokines, the signaling pathways followed by MSCs involved in the burn wound healing along with their factors. Cells signals constitute a very dynamic and promising research field, and may be a source contributing to burn scar contractures.

Authors response:

In this study Authors report that burn eschar pericytes, a subpopulation of mesenchymal stem cells (MSCs), are inflammatory and may contribute to burn scar contractures and fibrosis, representing a population of adult stem cells mobilized in response to the burn injury. Results from these studies provide clues to pericytes in the burn wound environment for better healing outcome in reducing scar contractures, their role in vessel formation and how they can affect the wound healing process.

1) Previous and even recent published articles, some of them not cited by the authors: Vincent C. van der Veen et al., Cell Transplantation, 2012; Popescu et al., Rom J Morphol Embryol 2011, Mills et al., Cells 2013 had reviewed the literature in this area. The authors should amplify their Bibliography including additional references (see above) to give additional information in the final interpretation of findings for our understanding the relationship of pericytes dissociated from vessels, their hyperactive state in response to the various factors secreted by activated immune cells and their differentiation into myofibroblasts promoting excessive fibroplasia.

We thank the reviewer for identifying more articles which we missed to cite it in the manuscript. We now have included citations recommended by the reviewer to the manuscript, which is relevant to our studies. We have added an additional 25 references to support our interpretation. These added references are highlighted in the text. We have provided additional information in the introduction on pericytes multipotency (lines 57-63, page 2). The discussion has been enhanced with the addition of a section that discusses pericyte dissociation from vessels and differentiation into myofibroblasts (lines 508-522; pages 15-16). We also discuss the role pro-inflammatory cytokines (TGF-β1, IL-1β, IL-6 and IL-8) play in burn wound healing and fibrosis (lines 536-547 and 555-565, 572-574; pages 16-17). We feel this additional information will provide the readers a greater understanding of how the pro-inflammatory cytokine observed in the burn eschar affects the wound environment and pericytes to promote fibrosis. We have interpreted our findings to the previous studies and provided our insights on our findings.

2) Authors should amplify their text, discussion and conclusions including information derived from other reported studies, to give additional information in the final interpretation of findings for understanding the the relationship between pericytes from burn wounds, expression levels of pro-inflammatory cytokines, the signaling pathways followed by MSCs involved in the burn wound healing along with their factors. Cells signals constitute a very dynamic and promising research field, and may be a source contributing to burn scar contractures.

As per the reviewer’s recommendations, we have amplified the text (lines 280-304, 354-358 and 367-370), discussion (506-520,528-529,534-545,552-563; pages 15-17, and 628-636; page 18) and conclusions (649-653; page 18) section of the manuscript. As the reviewer mentions the finding that pericytes are hyperactive by producing excessive amounts of pro-inflammatory cytokines and expresses myofibroblasts markers may have a significant role in burn scar contractures which requires further investigation. Our future studies will focus on the growth factors, in particular, TGF-β1 and its contribution to modulating pericytes function.

Reviewer 2 Report

The manuscript entitled "Characterization of Burn Eschar Pericytes" describes about impact of burn wound on pericytes during healing process. They  compared the gene and protein expression of pericytes isolated from normal skin and burn eschar tissues.The authors also reported the expression of unique transcription factor FOXE1 in normal skin pericytes, which is elevated in burn  eschar pericytes. Overall, outcome of finding is very interesting and valuable for the readers.

Queries:

Introduction: I suggest to include some recent findings about pericytes from pluripotent stem cells. I would like to see the expression of some of the smooth cells markers to make sure that isolated cells are exclusively pericytes and not smooth muscles.  Isolated pericytes are proliferative of need to isolate freshly for each experiments? Please explain with supporting data. I do not understand what authors want to say on the identification of the expression of FOXE1 transcription factor. It is a very interesting finding though. I would suggest working on writing to make it more relevant to the context and understandable to the readers. I would to see the morphology (Phase contrast)  of the isolated pericytes. 

Authors response:

The manuscript entitled “Characterization of Burn Eschar Pericytes” describes about impact of burn wound on pericytes during healing process. They compared the gene and protein expression of pericytes isolated from normal skin and burn eschar tissues. The authors also reported the expression of unique transcription factor FOXE1 in normal skin pericytes, which is elevated in burn eschar pericytes. Overall, outcome of finding is very interesting and valuable for the readers.

We are appreciative of the reviewer’s comment that the study is interesting and valuable for the readers.

Introduction: I suggest to include some recent findings about pericytes from pluripotent stem cells. I would like to see the expression of some of the smooth cells markers to make sure that isolated cells are exclusively pericytes and not smooth muscles. Isolated pericytes are proliferative of need to isolate freshly for each experiments? Please explain with supporting data. I do not understand what authors want to say on the identification of the expression of FOXE1 transcription factor. It is a very interesting finding though. I would suggest working on writing to make it more relevant to the context and understandable to the readers. I would to see the morphology (Phase contrast) of the isolated pericytes.

We have provided recent findings regarding pericytes pluripotency and pericyte detachment from the vessel seems to be one of the initiating factors in pericytes dedifferentiation into a pluripotent cell (lines 57-63; page 2, 283-288; page 7, and 508-522; pages 15-16).

As per the reviewer’s recommendation, we have used three different antibodies to stain for smooth muscle cell markers namely, SM22, Calponin-1, and MHY11, which are shown in Figure 2e. From our findings, we have inferred that some of the burn eschar-derived pericytes acquire smooth muscle cell gene expression similar to myofibroblasts which are due to their multipotential nature as suggested previously (Kumar et al. Specification and Diversification of Pericytes and Smooth Muscle Cells from Mesenchymoangioblasts. Cell Reports 2017 May 30; 19 (9):1902-1916.) Due to this plasticity in their cellular fate, we observed a significant number of cells that are positive for SM22 but not calponin-1 or MHY11. However, this again reiterates that these cells might still possess the mesenchymal stem cell-like property and can differentiate into various other cell populations showing their plasticity. If the reviewer is pointing to the proliferative nature of pericytes, we have shown it in Figure 7a. We found that burn eschar pericytes display a slight increase in the proliferation rate (though not statistically significant) compared to normal skin-derived fibroblasts and pericytes. The population doubling time for both cell populations was 24 hours.

If the reviewer is pointing to the proliferative nature of pericytes, we have shown it in Figure 7a. We found that burn eschar pericytes display a slight increase in the proliferation rate (though not statistically significant) compared to normal skin-derived fibroblasts and pericytes. The population doubling time for both cell populations was 24 hours. Frozen down primary cultures of pericytes were used until passage number 5.

We are also in agreement with the reviewer’s comment that the identification of FOXE1 is an interesting finding, and it warrants further investigation. FOXE1 gene was the top gene that was identified through RNA-seq analysis and interestingly; there was low to no expression of FOXEl in fibroblasts. FOXE1 can be a new marker to identify pericytes which can be unique and not seen in fibroblasts. Currently, we are exploring the relationship between FOXE1 and pericytes and the role played by FOXE1 in the skin. We have elaborated on our writing in the discussion section of the manuscript on our hypothesis on FOXE1 and previous studies on the other members of the FOX family (lines 630-638; page 18). We hope that the text in the current form is understandable and clear. Phase-contrast images of normal- and burn eschar-derived pericytes are included in Figure 1b.

Round 2

Reviewer 1 Report

Authors have adequately addressed comments raised in previous round of review.

Reviewer 2 Report

I have no further comments.